# Geographic inequities in human papillomavirus vaccine non-uptake and its determinants among adolescent girls in Ethiopia: Evidence from the National Immunization Survey

Kibir Temesgen Assefa[1,2]*, Abebaw Gebeyehu Worku[1], Achenef Asmamaw Muche[3], Bisrat Misganaw Geremew[3], Mulat Adefris Woldetsadik[1], Kassahun Alemu[3]

1 Department of Reproductive Health, Institute of Public Health, College of Medicine and Health Sciences, University of Gondar, Gondar, Ethiopia, 2 Department of Midwifery, College of Medicine and Health Sciences, Wollo University, Dessie, Ethiopia, 3 Department of Epidemiology and Biostatistics, Institute of Public Health, College of Medicine and Health Sciences, University of Gondar, Gondar, Ethiopia

* kibirwollo@gmail.com

## Abstract

### Introduction

Human papillomavirus (HPV) vaccination has emerged as the most effective method for preventing cervical cancer. Despite this, Ethiopia's HPV vaccine non-uptake rate remains high, with significant geographic variation, and there is limited evidence on the geospatial determinants of these inequities. This study aimed to map the geographic inequities in HPV vaccine non-uptake and identify its determinants among adolescent girls in Ethiopia.

### Methods

We conducted a secondary data analysis using the Ethiopian National Immunization Survey dataset. A stratified two-stage cluster sampling technique was used to select 467 enumeration areas (EAs) and a weighted sample of 5,341 adolescent girls. The geographic inequity of HPV vaccine non-uptake was analyzed using Moran's I, Getis-Ord Gi statistics, and Kriging interpolation in ArcGIS 10.8. We employed geographically weighted regression analysis to identify geographic factors associated with inequity in HPV vaccine non-uptake.

### Results

Forty-six percent (46%, 95% CI: 44.7–47.8) of adolescent girls did not receive the HPV vaccine, and there were geographical variations in vaccine coverage. Higher proportions of HPV vaccine non-uptake were identified in eastern Amhara, eastern Oromia, central and northern Somali, central Afar, and the urban administrative units of Dire Dawa and Harari. Poor attitude and poor knowledge towards the HPV

**Data availability statement:** All relevant data are within the manuscript and its Supporting Information file.

**Funding:** This work is part of PhD study for KTA and was funded by College of Medicine and Health Sciences, University of Gondar with reference number of PGC//098/12/2016. The funders had no role in study design, data collection and analysis, decision to publish, or preparation of the manuscript.

**Competing interests:** The authors have declared that no competing interests exist.

vaccine, not living with parents, and urban residence were predictors of geographic inequities in HPV vaccine non-uptake.

## Conclusions

The proportion of HPV vaccine non-uptake varied across Ethiopia, with geographic inequities identified in the eastern and northeastern parts of Ethiopia. Poor attitudes and knowledge about the vaccine, not living with parents, and urban residence contributed to these inequities. These findings highlight the need for targeted educational campaigns in areas with high non-uptake to improve knowledge and attitudes, alongside tailored strategies for regions where urban residence and not living with parents influence uptake.

---

## Introduction

Human papillomaviruses (HPVs) are a group of epitheliotropic viruses, many of which are carcinogenic and strongly associated with cervical, anal, oropharyngeal, and other anogenital cancers [1]. HPV-related cervical cancer is the most common cancer in women worldwide [2]. Globally, HPV infects 80% of sexually active individuals, with low- and middle-income countries accounting for more than 90% of cases [3]. In Ethiopia, 31.5 million women are at risk of contracting HPV, which causes 99% of cervical cancers, with over 7,445 new cases annually, and 70% of cases dying from the disease [4,5]. Over 90% of cervical cancers can be prevented with highly effective, safe, and free or low-cost HPV vaccines [6–9].

Despite the effectiveness and safety of HPV vaccines, the non-uptake rate remains high [10–13], and significant geographic inequities exist both between and within countries [14–16], ranging from 5.6% to 98.9% [17]. The 2023 WHO report estimated that globally, seven out of eight girls remain unvaccinated [17]. The Strategic Advisory Group of Experts Vaccine Hesitancy Determinants Matrix identified contextual, individual, and multi-level factors responsible for non-uptake and geographic inequities [18]. Socioeconomic status, including income, occupational status, educational level, residence, and geographical location, as well as vaccine access, contributed to HPV vaccine uptake [19–21]. Studies from Switzerland [22], the USA [23], China [24], Kenya [25], and Uganda [26] reported that local policies, community attitudes, socio-economic conditions, racial differences, culture, and healthcare access problems are significant factors for HPV vaccine non-uptake and coverage inequities.

The recent Ethiopian National Immunization Evaluation Survey reported regional variation in HPV vaccine non-uptake, ranging from 26.4% in Benishangul Gumuz to 67.2% in Somali [27]. These regional differences highlight the need to assess sub-national variations and factors associated with each specific geographic area to inform targeted and tailored interventions [27]. Although many studies worldwide have identified socio-demographic factors affecting HPV vaccine non-uptake [28–37], little is known about the geospatial determinants contributing to uptake inequities,

particularly in Ethiopia's geographic context. As far as our search is concerned, no study has assessed geographic inequities in HPV vaccine non-uptake and its determinants in Ethiopia. Mapping HPV vaccine coverage disparities is essential to support cervical cancer prevention efforts.

Therefore, this study aimed to determine geographic inequities in HPV vaccine non-uptake and identify its determinants among adolescent girls in Ethiopia.

## Materials and methods

### Study setting, design, and period

This study was a secondary analysis of a National Immunization Program Evaluation Survey conducted from March to July 2023 in the two city administrations and all regions of Ethiopia, except Tigray due to conflicts. The survey was conducted by the Ministry of Health in collaboration with Addis Ababa University, Jimma University, Haramaya University, Hawassa University, and the University of Gondar. In Ethiopia, HPV vaccination is delivered through schools, with community and health facility outreach for girls not enrolled in school. The Expanded Program on Immunization (EPI) provides HPV vaccines at no cost, and the Ministry of Health coordinates the service in collaboration with regional health bureaus. During data collection, HPV vaccination services were available across all regions of the country, although geographic and socioeconomic inequities in uptake persist. Detailed information about the study setting is described in the original study [27].

### Study population and sampling

The source population comprised all adolescent girls aged 15 years or older in Ethiopia, while the study population included those adolescent girls in the selected enumeration areas (EAs). However, our analysis included only girls aged 15–18 years. This was due to the nature of the secondary data, which were originally collected from adolescent girls eligible for HPV vaccination under a single-age cohort strategy targeting only 14-year-olds. As data were collected retrospectively, one year after the vaccination campaign, participants who were 14 at the time of vaccination had turned 15 by the time of data collection. Therefore, 15 years was used as the lower age limit, while the upper age limit (18 years) was determined by the maximum age observed in the dataset. The sample size was estimated using the single population proportion formula at both national and regional levels. A stratified two-stage cluster sampling technique was used to select 467 EAs and a weighted sample of 5,341 adolescent girls. From each EA, which consisted of 180–200 households, 30 households with at least one eligible participant were randomly selected. EAs and households were selected as the primary and secondary sampling units, respectively. EAs were randomly selected from urban and rural areas using the Ethiopian Statistical Services (ESS) sampling frame, with the number of EAs per region or city determined according to their estimated sample sizes. The detailed sampling procedure was published elsewhere [27].

### Data sources, data collection tools, and procedures

We obtained data on HPV vaccination status, explanatory variables, and geographic coordinates from the 2024 National Immunization Evaluation Survey adolescent dataset. We accessed the dataset on 26 November 2024. We did not have access to information that could identify individual participants during or after data collection, as all data were fully anonymized. The outcome variable was geo-referenced at the EA level and linked to area-level covariates using ArcGIS. We obtained administrative boundary or polygon shapefiles of Ethiopia from the platform https://data.humdata.org/dataset/cod-ab-eth, which provides open-access data publicly available for unrestricted use. Household location data were collected using global positioning system (GPS) coordinates (latitude and longitude) after each interview with eligible participants. The survey locations were represented by points. To maintain confidentiality, EA centroids were randomly displaced

in direction (0–360°) and distance (up to 2 km in urban and 5 km in rural settings, due to differences in population density), according to the Ethiopian Demographic and Health Survey (EDHS) protocol.

Pretested questionnaires were used to collect data through face-to-face interviews. The questionnaires were translated into five local languages (Afar, Amharic, Afaan Oromoo, Sidama, and Somali). Data collectors and supervisors were recruited from the local areas, and seven days of training were provided to them. Vaccination cards and health facility records were used to cross-check vaccination status, in addition to respondents' recall.

## Measurements

**Outcome variable.** The outcome variable in our study is the non-uptake of the HPV vaccine. An adolescent girl aged 15 years or older was considered not vaccinated (i.e., non-uptake of the HPV vaccine) if she had not received any dose of the HPV vaccine at the time of data collection [27]. We define geographical inequity in HPV vaccine non-uptake as certain EAs having more adolescent girls who did not receive the vaccine than others. We identified EAs with geographical inequity (high non-uptake of the HPV vaccine) using hotspot analysis and spatial interpolation techniques.

**Explanatory variables.** The explanatory variables were age of adolescent girls, literacy, current school attendance, religion, living arrangements, maternal education, paternal education, maternal occupation, paternal occupation, wealth index, residence, knowledge of HPV, cervical cancer and HPV vaccine, and attitude towards the HPV vaccine. The household wealth index was constructed using principal component analysis (PCA) [27]. The index incorporated housing characteristics (flooring, roofing, wall materials, water source, toilet facility, and cooking fuel) and asset ownership (television, radio, refrigerator, bicycle, motorcycle, car, mobile telephone, agricultural land, and livestock). PCA generated factor scores, with the first principal component used to create the wealth index, ranking households into quintiles from poorest to wealthiest. For our analysis, we categorized households into five wealth levels (poorest, poorer, middle, richer, and richest), following standard Demographic and Health Survey (DHS) practice to enable cross-study comparisons.

## Geographic equity analysis

We presented the socio-demographic characteristics of adolescent girls aged 15 years or older and HPV vaccine non-uptake using descriptive statistics with 95% confidence intervals. To ensure national representativeness of the included adolescent girls, we adjusted our analyses for sampling weights. To account for the two-stage sampling design, we calculated sampling weights as the inverse probability of selecting each enumeration area and household.

We analyzed the spatial distribution of HPV vaccine non-uptake among adolescent girls using ArcGIS 10.8 software. We assessed the spatial autocorrelation of HPV vaccine non-uptake using Global Moran's I statistic to determine whether it was clustered, dispersed, or randomly distributed across enumeration areas. A Moran's I value close to zero indicated a random pattern, while values near +1 and –1 suggested clustering and dispersion, respectively. A high Z-score ($z > 1.96$) and low p-value ($p < 0.05$) provided evidence of statistically significant clustering and indicated a meaningful geographic trend. To identify areas with high or low non-uptake, we conducted a hotspot analysis using the Getis-Ord Gi* statistic, where positive z-scores indicated hotspots (areas with high non-uptake) and negative z-scores indicated cold spots (areas with low non-uptake). We used ordinary kriging interpolation to predict the proportion of HPV vaccine non-uptake in unsampled areas. This method was selected because it produced the lowest mean prediction error and root mean square prediction error, making it the most reliable interpolation technique for the dataset [38].

To identify the determinants of geographic variation in HPV vaccine non-uptake, we first conducted an ordinary least squares (OLS) analysis. The OLS model assumptions were evaluated using the Joint Wald test for overall model significance ($p < 0.05$), the Jarque-Bera test for normality of residuals ($p > 0.05$), the Koenker (BP) statistic for stationarity of relationships ($p > 0.05$), and the variance inflation factor ($VIF < 7.5$) for multicollinearity. Among these assumptions, spatial stationarity between the outcome and explanatory variables was violated, as indicated by a significant Koenker (BP) test ($p < 0.05$), justifying the use of geographically weighted regression (GWR) as the best-fitting model. Accordingly, predictor

variables with p < 0.05 were included in the GWR model, which allows parameter estimates to vary by location and capture local influences on HPV vaccine non-uptake. Model performance between OLS and GWR was also compared using the corrected Akaike Information Criterion (AICc) and adjusted R², with GWR identified as the best-fitting model due to the lowest AICc and highest adjusted R² [39].

### Ethical approval

Ethical approval was granted by the College of Medicine and Health Sciences, University of Gondar Institutional Review Board (Ref. No: CMHSSH-UOG IRERC/43/05/2024). These secondary data were collected in accordance with the Helsinki Declaration. Information that could identify individual participants was fully anonymized. For geographic data, coordinates representing enumeration areas (EAs) were intentionally displaced in direction (0–360°) and distance (2 km for urban areas and 5 km for rural areas) to protect participant confidentiality by preventing identification of specific locations.

## Results

### Socio-demographic characteristics of study participants

Of the weighted sample of 5,341 adolescent girls, 61% were aged 15–16 years (mean age 16.24 ± 1.1). Most participants lived in rural areas (75.9%), with the largest representation from Oromia (40.1%) and Amhara (22.9%) regions. Overall, 85.1% could read and write, and 63.8% were attending school. Orthodox Christianity (47.3%) and Islam (35.7%) were the most common religions (Table 1).

### National and regional human papillomavirus vaccine non-uptake

The proportion of HPV vaccine non-uptake among adolescent girls in Ethiopia was 46% (95% CI: 44.7–47.8). Significant geographical inequities were observed, with the highest non-uptake in Somali at 67.2% (95% CI: 62.7–71.4) and the lowest in Benishangul Gumuz at 26.4% (95% CI: 21.1–32.6) (Fig 1).

### Geographical autocorrelation of human papillomavirus vaccine non-uptake

HPV vaccine coverage in Ethiopia showed notable geographical inequities, with spatial autocorrelation analysis indicating a positive and statistically significant clustering of non-uptake (Global Moran's I = 0.87, Z = 3.2, P < 0.01) (Fig 2).

### Geographical patterns (hotspot and cold spot areas) of human papillomavirus vaccine non-uptake

The highest proportion of HPV vaccine non-uptake was found in the border areas of eastern Amhara, eastern Oromia, central and northern Somali, central Afar, Dire Dawa, and Harari, while the lowest non-uptake was concentrated in central and northern Southwest Ethiopia, northern Sidama, northern SNNPR, central Gambela, and central and western Benishangul-Gumuz (Fig 3).

### Spatial interpolation of human papillomavirus vaccine non-uptake

Based on the observed data points from sampled areas, we estimated the proportion of HPV vaccine non-uptake in unsampled areas using Ordinary and Universal Kriging interpolation methods. These approaches provided the best fit, with the lowest mean predicted error (MPE = 0.00234) and root mean square predicted error (RMSE = 0.24908) (Table 2). Higher predicted proportions of non-uptake were mainly observed in most of the Somali region – particularly in the central and eastern zones – as well as in northern Afar, eastern Dire Dawa, eastern Harari, eastern Tigray, and parts of central and eastern Oromia. In contrast, lower predicted non-uptake was found in central Amhara, Benishangul-Gumuz, western Oromia, Gambella, central and northern SNNPR, northwestern Southwest Ethiopia, and the western part of the Somali region (Fig 4).

**Table 1. Socio-demographic characteristics of adolescent girls in Ethiopia, 2024 (n = 5,341).**

| Variables | Category | Unweighted (n = 5,336) Frequency (%) | Weighted (n = 5,341) (%) | HPV vaccination status | |
| --- | --- | --- | --- | --- | --- |
| | | | | Vaccinated (n = 2,873, 53.8%) Frequency (%) | Not vaccinated (n = 2,468, 46.2%) Frequency (%) |
| **Age in years** (Mean ± SD = 16.24 ± 1.1) | 15–16 | 3,203 (60) | 60.8 | 1,846 (64) | 1,399 (57) |
| | 17–18 | 2,133 (40) | 39.2 | 1,027 (36) | 1,069 (43) |
| **Able to read and write** | Yes | 4,557 (85.4) | 85.1 | 2,754 (95.9) | 1,790 (72.5) |
| | No | 779 (14.6) | 14.9 | 119 (4.1) | 678 (27.5) |
| **Educational status** | No formal education | 812 (15.2) | 15.8 | 155 (5.4) | 686 (27.8) |
| | Primary | 3,621 (67.9) | 67.4 | 2,176 (75.7) | 1,440 (80.8) |
| | Secondary | 903 (16.9) | 16.8 | 542 (18.9) | 342 (19.2) |
| **Currently attending school** | Yes | 3,458 (64.8) | 63.8 | 2,302 (80.1) | 1,109 (44.9) |
| | No | 1,878 (35.2) | 36.2 | 572 (19.9) | 1,359 (55.1) |
| **Religion** | Orthodox | 2,441 (45.7) | 47.3 | 1,175 (40.9) | 1,353 (54.8) |
| | Muslim | 1,679 (31.5) | 35.7 | 1,110 (38.6) | 796 (32.3) |
| | Protestant | 1,189 (22.3) | 16.7 | 578 (20.1) | 312 (12.6) |
| | Other* | 27 (0.5) | 0.3 | 10 (0.4) | 7 (0.3) |
| **Maternal education** | No formal education | 3,860 (72.3) | 73.8 | 1,977 (68.8) | 1,963 (79.6) |
| | Primary education | 956 (17.9) | 18.0 | 626 (21.8) | 335 (13.6) |
| | Secondary education | 283 (5.3) | 4.6 | 163 (5.7) | 82 (3.3) |
| | College and above | 139 (2.6) | 1.9 | 66 (2.3) | 39 (1.6) |
| | Don't know | 98 (1.8) | 1.7 | 41 (1.4) | 49 (1.9) |
| **Father's education** | No formal education | 3,030 (56.8) | 58.1 | 1,452 (50.5) | 1,649 (66.8) |
| | Primary education | 1,373 (25.7) | 26.0 | 871 (30.3) | 520 (21.1) |
| | Secondary education | 545 (10.2) | 9.3 | 342 (11.9) | 154 (6.2) |
| | College and above | 238 (4.5) | 3.7 | 134 (4.7) | 63 (2.6) |
| | Don't know | 150 (2.8) | 2.9 | 74 (2.6) | 82 (3.3) |
| **Maternal occupation** | House wife | 3,363 (63) | 63.7 | 1,816 (63.2) | 1,588 (64.3) |
| | Farmer/pastoralist | 1,314 (24.6) | 25.2 | 707 (24.6) | 637 (25.8) |
| | Merchant | 327 (6.1) | 5.8 | 186 (6.5) | 125 (5.1) |
| | Employed | 245 (4.6) | 3.6 | 115 (3.9) | 79 (3.2) |
| | Others*** | 87 (1.6) | 1.7 | 49 (1.7) | 39 (1.6) |
| **Father's occupation** | Farmer/pastoralist | 3,776 (70.8) | 73.3 | 2,073 (72.1) | 1,842 (74.7) |
| | Merchant | 528 (9.9) | 9.5 | 305 (10.6) | 203 (8.2) |
| | Employed | 634 (11.9) | 10.3 | 333 (11.6) | 219 (8.9) |
| | Daily labourer | 47 (0.9) | 0.8 | 20 (0.71) | 23 (0.92) |
| | Others**** | 351 (6.6) | 6.1 | 142 (4.9) | 181 (7.3) |

*(Continued)*

**Table 1.** (Continued)

| Variables | Category | Unweighted (n = 5,336) Frequency (%) | Weighted (n = 5,341) (%) | HPV vaccination status | |
|---|---|---|---|---|---|
| | | | | Vaccinated (n = 2,873, 53.8%) Frequency (%) | Not vaccinated (n = 2,468, 46.2%) Frequency (%) |
| **Living with** | Parents | 4,289 (80.4) | 81.4 | 2,477 (86.2) | 1,872 (75.9) |
| | Relatives | 737 (13.8) | 13.4 | 315 (10.9) | 400 (16.2) |
| | Husband | 156 (2.9) | 2.7 | 40 (1.4) | 103 (4.2) |
| | Others***** | 154 (2.9) | 2.5 | 41 (1.4) | 93 (3.8) |
| **Wealth index** | Poorest | 917 (17.2) | 17.0 | 364 (12.7) | 544 (22.0) |
| | Poor | 816 (15.3) | 15.1 | 383 (13.3) | 421 (17.1) |
| | Middle | 1,810 (33.9) | 35.8 | 1,026 (35.7) | 888 (35.9) |
| | Rich | 845 (15.8) | 15.9 | 533 (18.6) | 317 (12.8) |
| | Richest | 948 (17.8) | 16.2 | 567 (19.7) | 298 (12.1) |
| **Region** | Addis Ababa | 325 (6.1) | 3.9 | 90 (3.1) | 118 (4.8) |
| | Afar | 354 (6.6) | 2.0 | 36 (1.3) | 73 (2.9) |
| | Amhara | 831 (15.6) | 22.9 | 721 (25) | 502 (20.3) |
| | Benishan-gul Gumuz | 227 (4.3) | 1.2 | 48 (1.7) | 17 (0.6) |
| | Dire Dawa | 196 (3.7) | 0.5 | 12 (0.4) | 17 (0.7) |
| | Gambela | 210 (3.9) | 0.5 | 20 (0.7) | 8 (0.3) |
| | Hareri | 223 (4.2) | 0.2 | 6 (0.2) | 9 (0.3) |
| | Oromia | 1,315 (24.6) | 40.1 | 1,194 (41.6) | 945 (38.3) |
| | SNNPR | 581 (10.9) | 4.5 | 148 (5.2) | 92 (3.7) |
| | Sidama | 343 (6.4) | 6.6 | 229 (7.9) | 122 (4.9) |
| | Somali | 451 (8.5) | 13.6 | 238 (8.3) | 488 (19.8) |
| | Southwest Ethiopia | 280 (5.3) | 3.9 | 131 (4.6) | 77 (3.1) |
| **Residence** | Urban | 1,411 (26.4) | 24.1 | 707 (24.6) | 578 (23.4) |
| | Rural | 3,925 (73.6) | 75.9 | 2,166 (75.4) | 1,890 (76.6) |

Others* = Catholic, Adventist, Apostolic, Hawariyat, Mulu wengel, Lutiran, Kalicha, Wakefata and No religion; **Other** = Home servant, alone, Coworker, Daughter in law; **Others*** = daily labourer, student, Private work, Begging, No occupation, Handcrafts, Prostitution; **Others**** = Retired/not working, Begging, Religious leader, Construction, Student, Driver, military; **Others***** = Servant/housemaid, not relative, employer, family, aunt, no relationship, co-worker, brother in law; **SNNPR** = Southern nations, nationalities and peoples' region

## Determinants of the geographical inequities in human papillomavirus vaccine non-uptake in Ethiopia

Geographically weighted regression (GWR) was identified as a better model than ordinary least squares (OLS), with the lowest corrected Akaike's Information Criterion (AICc) and higher adjusted R² values. In the GWR analysis, poor attitude, poor knowledge, not living with parents, and urban residence were identified as significant predictors, together explaining 55.1% of the geographical variation in HPV vaccine non-uptake (Tables 3 and 4). The effects of these determinants showed substantial spatial variation, with local coefficients ranging from −9.38 to 12.71 for poor attitude, −7.79 to 10.51 for poor knowledge, −122 to 27.13 for not living with parents, and −0.13 to 0.11 for urban residence (Fig 5–8).

## Discussion

This study assessed geographic inequity and determinants of HPV vaccine non-uptake among adolescent girls in Ethiopia. The findings showed significant geographic inequity in HPV vaccine non-uptake across Ethiopian

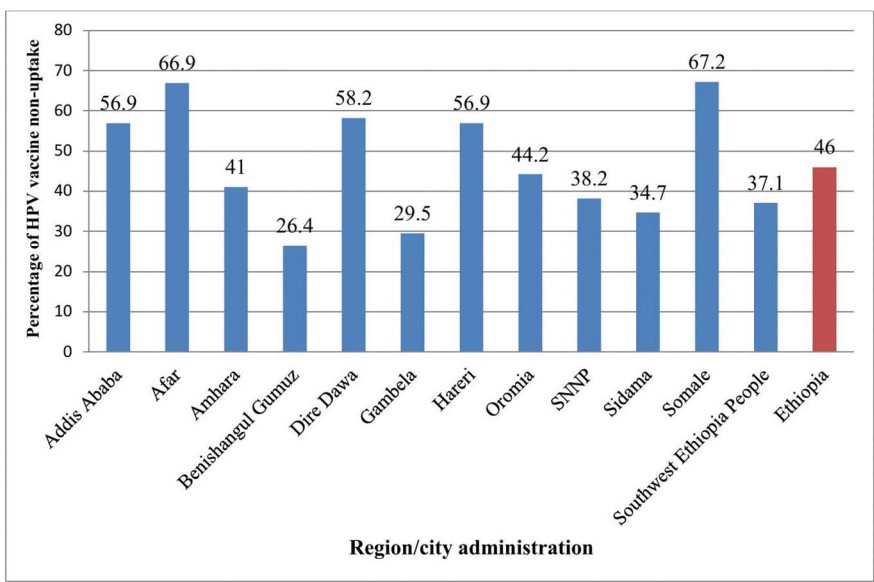

**Fig 1. Non-uptake of human papillomavirus (HPV) vaccine among adolescent girls in Ethiopia, 2024.** The figure shows the proportion of adolescent girls aged 15–18 years who did not receive any dose of the HPV vaccine across regions of Ethiopia. SNNP: Southern Nations, Nationalities, and Peoples.

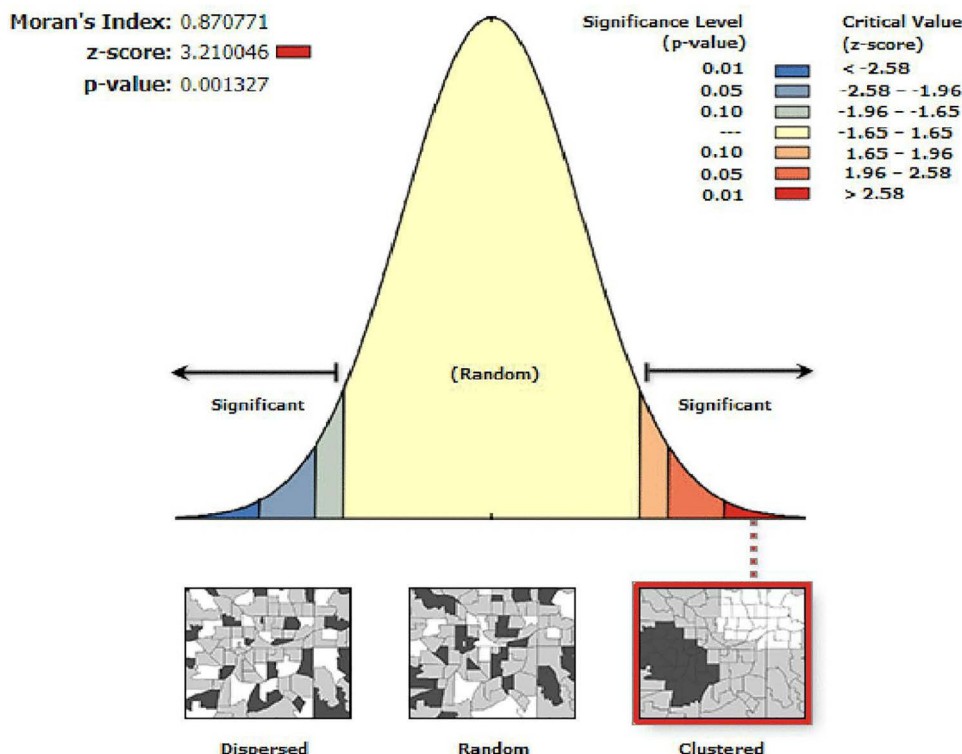

**Fig 2. Spatial autocorrelation of human papillomavirus (HPV) vaccine non-uptake among adolescent girls in Ethiopia, 2024.** Global Moran's I statistics indicate the spatial clustering pattern of HPV vaccine non-uptake among adolescent girls across Ethiopia.

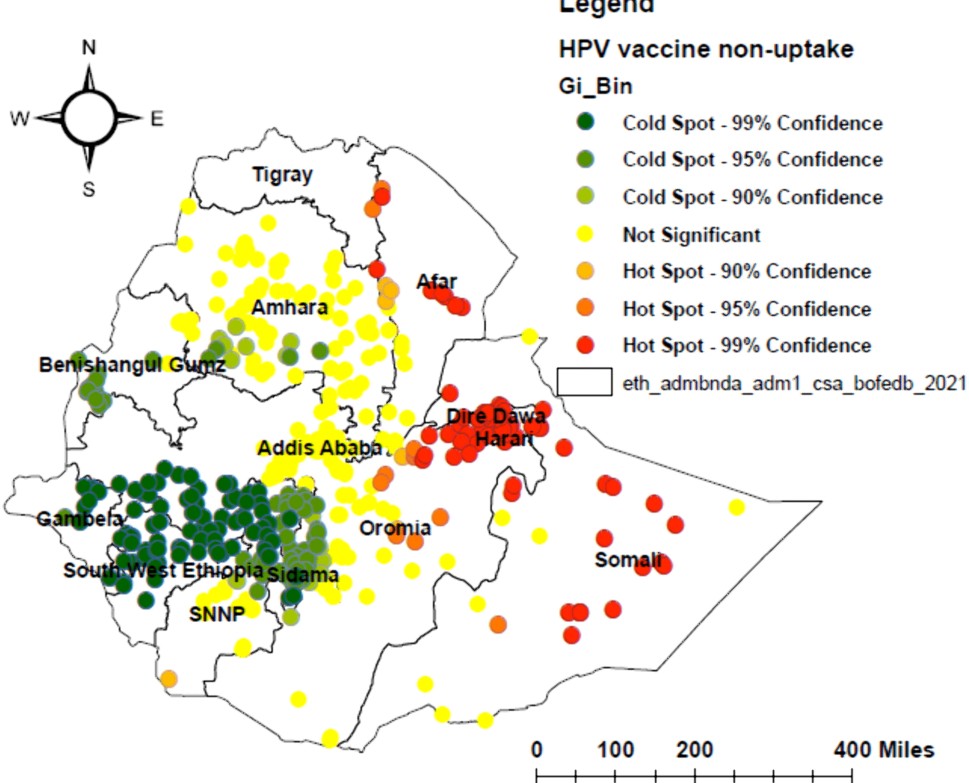

**Fig 3. Hotspot areas of human papillomavirus (HPV) vaccine non-uptake among adolescent girls in Ethiopia, 2024.** The map displays geographic hotspot and coldspot areas of HPV vaccine non-uptake identified using spatial autocorrelation analysis.

Table 2. Comparison of interpolation methods for human papillomavirus vaccine non-uptake among adolescent girls in Ethiopia, 2024.

| Interpolation method | Parameter | |
|---|---|---|
| | **Mean error (ME)** | **Root-mean-square error (RMSE)** |
| **Deterministic interpolation method** | | |
| Inverse distance weighted (IDW) | 0.00263 | 0.25909 |
| **Geostatistical interpolation methods** | | |
| **Ordinary kriging** | **0.00234** | **0.24908** |
| Simple kriging | 0.00872 | 0.26686 |
| **Universal kriging** | **0.00234** | **0.24908** |
| Disjunctive kriging | 0.00875 | 0.26686 |
| Probability kriging | −0.00022 | 0.48085 |
| Indictor kriging | 0.00368 | 0.49851 |

regions. The non-uptake rate was 46%, ranging from 26.4% in Benishangul-Gumuz to 67.2% in Somali. Factors associated with this geographic inequity included poor attitude, poor knowledge, not living with parents, and urban residence. The Federal Ministry of Health should allocate resources to areas with high non-uptake

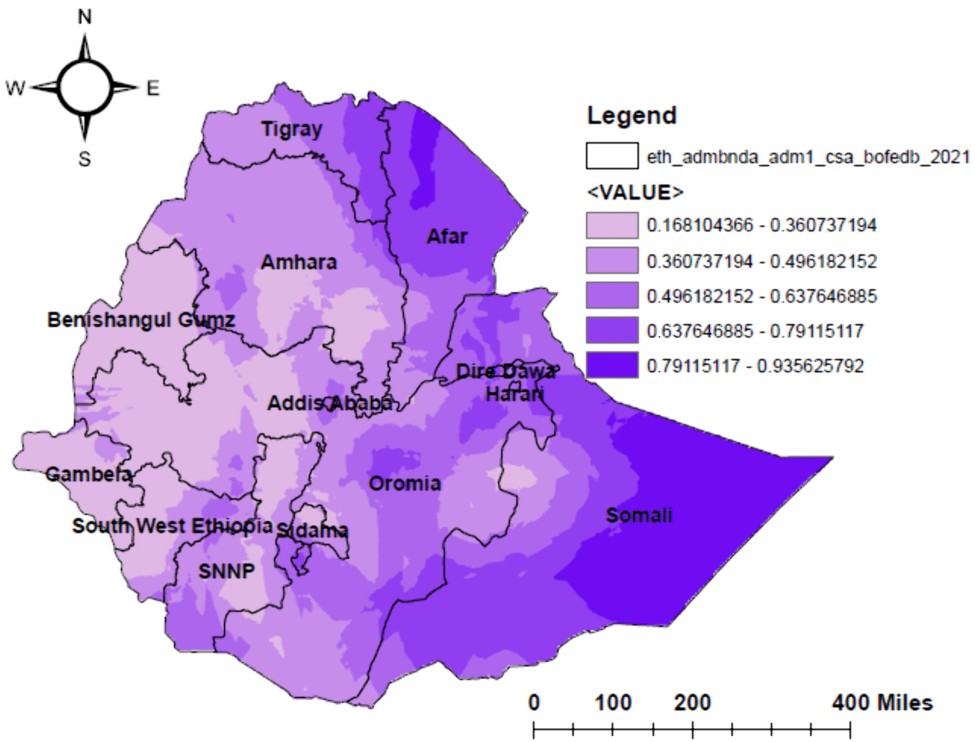

## Kriging interpolation of HPV vaccine non-uptake

**Legend**

☐ eth_admbnda_adm1_csa_bofedb_2021

**\<VALUE\>**

| | |
|---|---|
| | 0.168104366 - 0.360737194 |
| | 0.360737194 - 0.496182152 |
| | 0.496182152 - 0.637646885 |
| | 0.637646885 - 0.79115117 |
| | 0.79115117 - 0.935625792 |

**Fig 4. The predicted geospatial map for non-uptake of human papillomavirus vaccine among adolescent girls in Ethiopia, 2024.**

**Table 3. Global beta coefficients of the ordinary least squares model summary and diagnostics for human papillomavirus vaccine non-uptake among adolescent girls in Ethiopia, 2024.**

| Variable | Coefficient | Std. error | Probability | Robust probability | VIF |
|---|---|---|---|---|---|
| Intercept | 0.087196 | 0.100984 | 0.388320 | 0.435291 | ---- |
| Urban residence | 0.058515 | 0.023950 | 0.014921* | 0.030269 | 1.517 |
| Not able to read and write | 0.164154 | 0.097849 | 0.094115 | 0.107681 | 5.416 |
| Not attended school | −0.180438 | 0.093758 | 0.054907 | 0.081455 | 5.401 |
| Poor attitude | 0.333403 | 0.044272 | **0.000000*** | **0.000000*** | 1.617 |
| Poor knowledge | 0.322652 | 0.039233 | **0.000000*** | **0.000000*** | 1.485 |
| No paternal formal education | 0.037557 | 0.038549 | 0.330432 | 0.311677 | 1.503 |
| Not living with parents | 0.163254 | 0.049054 | **0.000959*** | **0.001628*** | 1.427 |
| Not currently attending school | 0.025957 | 0.050014 | 0.604029 | 0.608877 | 2.489 |
| **Ordinary least square (OLS) diagnostics** | | | | | |
| Diagnostic criteria | Magnitude | | p-value | | |
| AICc | −246.473402 | | | | |
| Multiple R squared | 0.536055 | | | | |
| Adjusted R squared | 0.527951 | | | | |
| Joint F-Statistics | 66.148277 | | 0.00000* | | |
| Joint Wald Statistics | 815.781436 | | 0.00000* | | |
| Koenker (BP) Statistics | 43.976105 | | 0.000001* | | |
| Jarque-Bera Statistics | 0.148994 | | 0.928210 | | |

**Table 4. Geographically weighted regression (GWR) analysis of human papillomavirus vaccine non-uptake among adolescent girls in Ethiopia, 2024.**

| Explanatory variable | poor attitude, poor knowledge, not living with parents, living in urban |
|---|---|
| Residual square | 13.71 |
| Effective number | 39.73 |
| Sigma | 0.179 |
| AICc | −259.279 |
| Multiple R square | 0.588 |
| Adjusted R square | 0.551 |

NB: AICc: Akaike's Information Criterion corrected

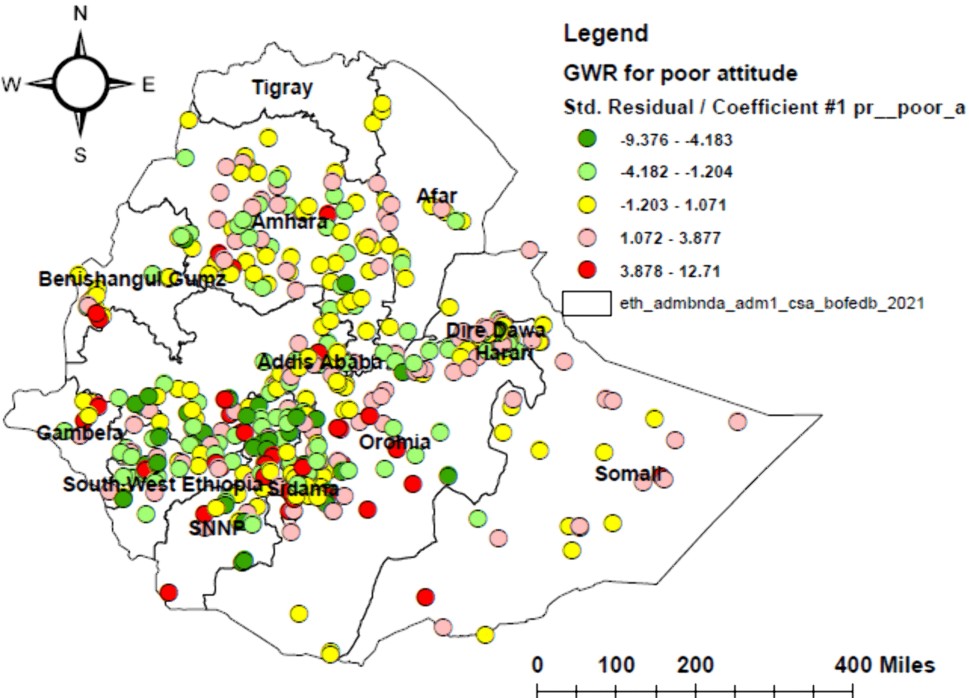

**Fig 5. Geographically weighted regression (GWR) map of poor attitude among adolescent girls in Ethiopia, 2024.** The map displays the spatial variation in the association between poor attitude and HPV vaccine non-uptake among adolescent girls.

and mandate the development of context-specific delivery strategies to address geographic inequities in HPV vaccination.

A high proportion of non-uptake of the HPV vaccine was observed in the eastern, southeastern, and northeastern parts of Ethiopia. This may be due to lower socioeconomic status, reduced healthcare-seeking behavior, and limited access to health services in these hard-to-reach areas, consistent with earlier studies [40,41]. These conditions may reinforce widespread misconceptions about vaccines and further reduce uptake. In contrast, low proportions of non-uptake were seen in central, northwestern, and southwestern areas, likely due to better infrastructure and stronger access to healthcare, in line with previous studies [42–44]. From a policy perspective, this spatial pattern indicates that a uniform national vaccination

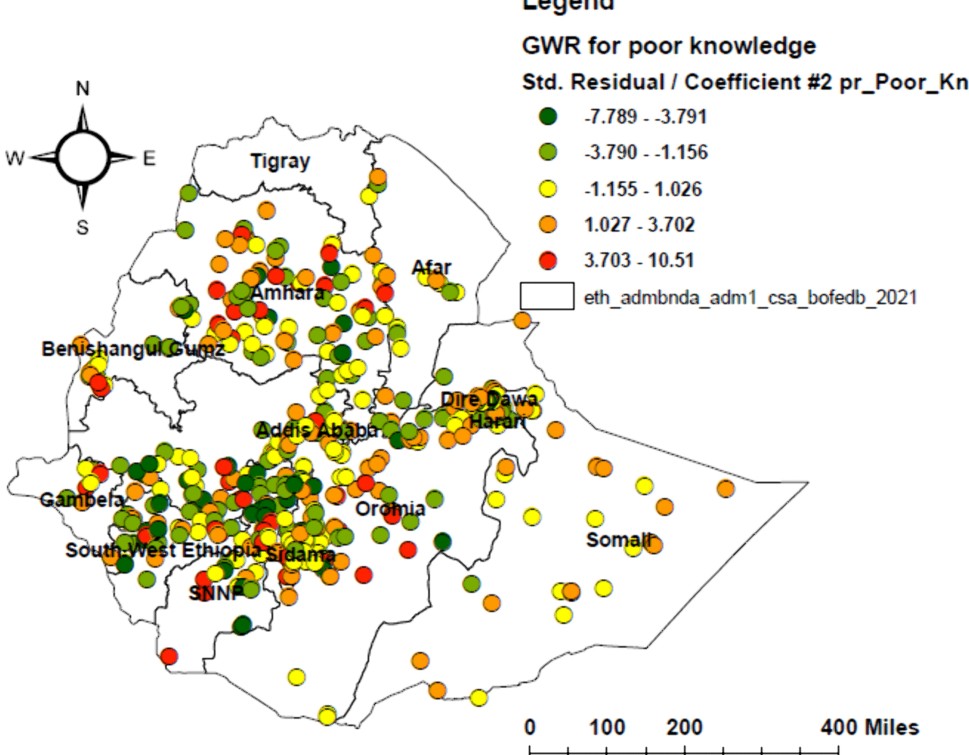

**Fig 6. Geographically weighted regression (GWR) map of poor knowledge among adolescent girls in Ethiopia, 2024.** The map shows the spatial variation in the relationship between poor knowledge and HPV vaccine non-uptake among adolescent girls.

strategy is inadequate. HPV vaccination programs in Ethiopia should therefore adapt geographically targeted approaches, including outreach services aligned with seasonal migration and integration with existing interventions such as mobile health and nutrition services, particularly in pastoralist and hard-to-reach regions like Afar and Somali.

Our study revealed that a poor attitude towards the HPV vaccine was significantly positively associated with non-uptake in central and southern Somali, central and eastern Oromia, central and eastern Amhara, northern Gambella, southern Benishangul-Gumuz, southern SNNPR, Dire Dawa, and Harari. This may be due to vaccine hesitancy, misinformation, and cultural resistance. In contrast, in central and northern Southwest Ethiopia, northern and central SNNPR, northern and central Benishangul-Gumuz, and central and western Oromia, a poor attitude had little influence on non-uptake, possibly due to strong community mobilization and supportive cultural norms. Our finding is consistent with studies conducted in Uganda [45], Kenya [46], and Tanzania [47]. The observed spatial variation in the effect of poor attitude on HPV vaccine non-uptake may be attributed to differences in cultural or religious beliefs, socioeconomic disparities, and limited access to trusted information sources. These patterns suggest that public health policies should not only focus on sharing information but also support the co-design of geographically tailored health education and communication strategies. These efforts should involve trusted community figures, such as elders and religious leaders, to address local misinformation and build trust.

Poor knowledge of HPV, cervical cancer, and the HPV vaccine was a strong positive predictor of non-uptake in Dire Dawa, Harari, eastern and central Oromia, central Amhara, eastern and southern Somali, southeastern Sidama, central and northern SNNPR, and southern Benishangul-Gumuz. These areas may have limited access to accurate health information and weak community outreach, making knowledge gaps a significant barrier. Conversely, in central and northern

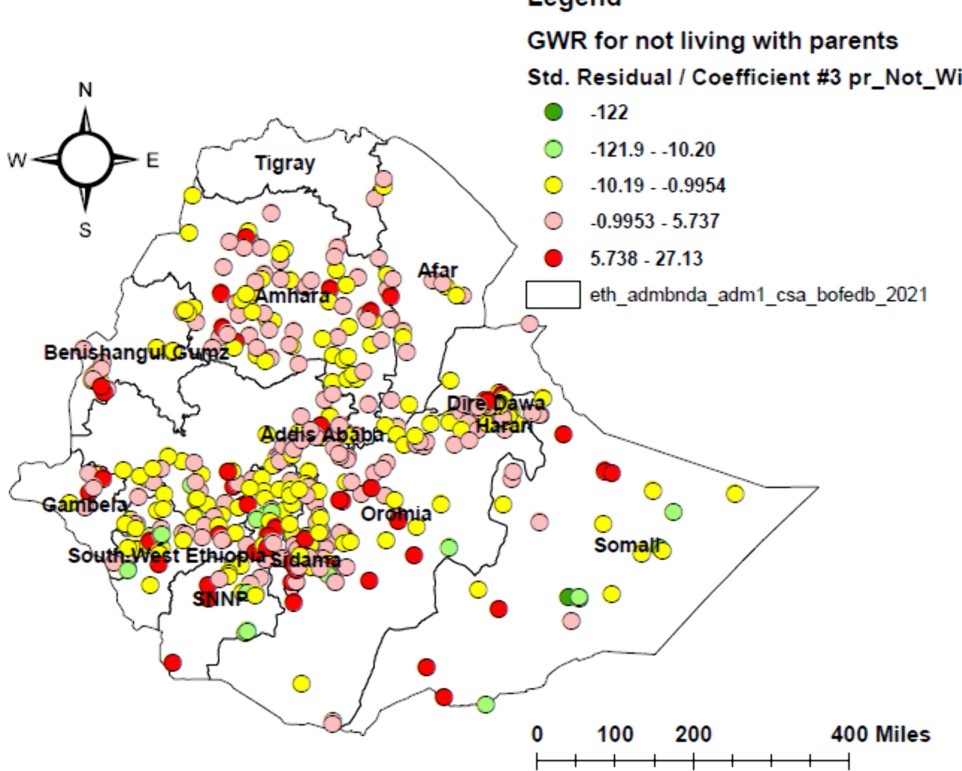

**Fig 7. Geographically weighted regression (GWR) map of not living with parents among adolescent girls in Ethiopia, 2024.** The map illustrates the spatial variation in the association between not living with parents and the HPV vaccine non-uptake among adolescent girls.

Southwest Ethiopia, central Gambela, northern and eastern Benishangul-Gumuz, northern SNNPR, and central and north-west Amhara, poor knowledge had little influence on vaccine non-uptake. In these areas, strong community engagement and trust in health workers may compensate for knowledge gaps. This finding is consistent with earlier studies conducted in Saudi Arabia [28], Kenya [48], Tanzania [49], and Nigeria [34]. These results highlight the need for geographically tailored health education integrated into health extension programs and school curricula to address misinformation and disparities. Policy efforts should prioritize training Health Extension Workers to deliver simple, standardized, and culturally appropriate messages about HPV prevention.

Our findings showed that not living with parents is positively and significantly associated with HPV vaccine non-uptake in Addis Ababa, Dire Dawa, southeastern Oromia, northeastern and southern Somali, central and eastern Amhara, central and northern SNNPR, southern Sidama, southern Benishangul Gumuz, central and northern Gambella, and central South West Ethiopia. This finding is consistent with studies conducted in India [50], Romania [51], and Uganda [36]. This association may result from the parents' role in providing consent and advocating for their children's immunization in urban and semi-urban areas. In contrast, the association between not living with parents and HPV vaccine non-uptake appears weak in northern and southern SNNPR, central Somali, and central South West Ethiopia, in line with other studies [52,53]. This may be because extended families, community elders, and school or religious leaders in rural areas can provide consent in the absence of parents. This finding highlights the need for policy makers to develop alternative consent mechanisms, such as mature minor consent or consent through school principals, in areas where not living with parents is a barrier to HPV vaccine uptake.

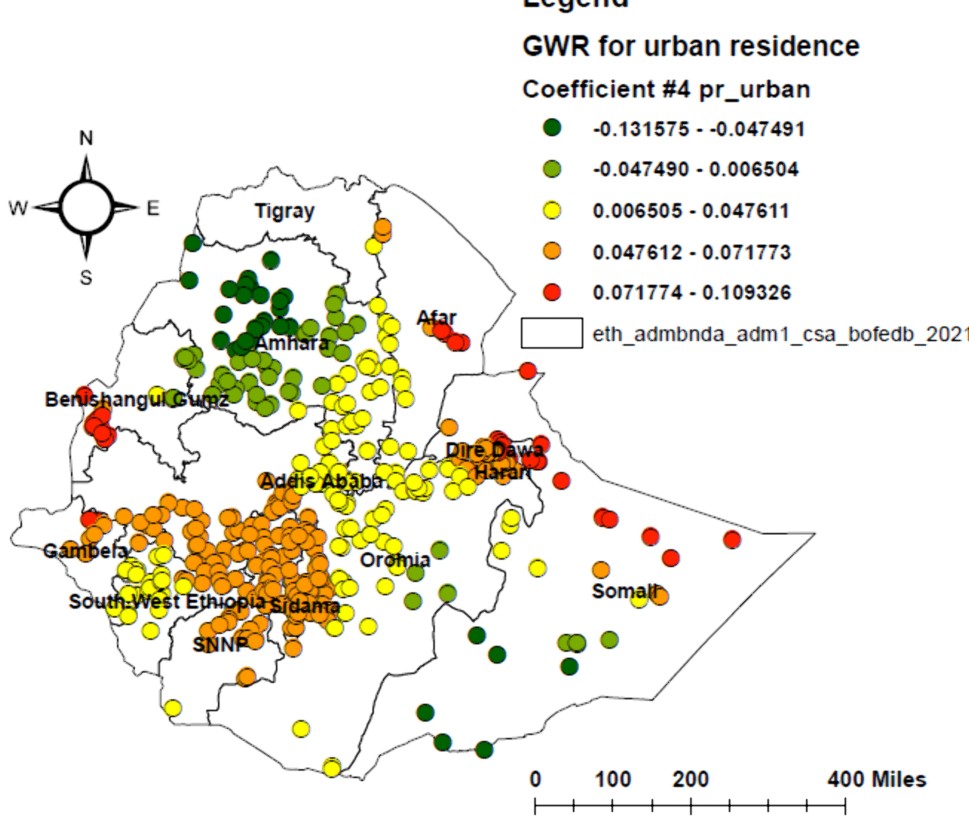

**Fig 8. Geographically weighted regression (GWR) map of urban residence among adolescent girls in Ethiopia, 2024.**

Urban residence also showed a positive and significant association with HPV vaccine non-uptake in central Afar, northeastern Somali, central and northern Gambella, central and northern SNNPR, central and northern Sidama, central and western Oromia, and western Benishangul Gumuz. This is consistent with studies in Uganda [54], Kenya [55], and Nigeria [56]. This highlights that urban settings do not always guarantee better vaccine uptake, due to factors such as urban poverty and informal fees [54], vaccine mistrust among urban elites [55], and fragmented health services in urban slums [56]. However, in northern and central Amhara, eastern Oromia, and southern Somali, urban residence is associated with higher HPV vaccine uptake, consistent with a study in Rwanda [57]. This may be due to better healthcare access and infrastructure in these urban areas. These findings suggest that urban vaccination policies should consider within-city heterogeneity and design targeted strategies for specific urban sub-populations, such as slum dwellers and low-income urban communities.

### Strengths and limitations of the study

This study used nationally representative data and advanced spatial modelling to provide robust national-level estimates and identify areas with high or low HPV vaccine non-uptake. Weighting was applied to increase the representativeness of the samples. The use of local data collectors familiar with the community likely reduced social desirability bias. Although data collectors attempted to cross-check data using vaccination cards and health facility records, reliance on self-reported vaccination history may introduce recall bias, especially when vaccination cards were unavailable.

## Conclusion

This study revealed substantial geographic inequity in HPV vaccine non-uptake across Ethiopia, indicating that tailored geographic interventions should be considered. HPV vaccine non-uptake was particularly clustered in the eastern and north-eastern parts of Ethiopia. Factors such as poor attitude, poor knowledge, not living with parents, and urban residence were associated with high non-uptake of the HPV vaccine. This study provides key insights for policymakers to develop tailored interventions, including enhancing education campaigns to address misconceptions, bridging knowledge and attitude gaps, developing alternative consent mechanisms such as self-consent (mature minor), and implementing community-based vaccination to overcome consent-related barriers.

## Supporting information

**S1 Dataset. Dataset of Geographic inequities in human papillomavirus vaccine non-uptake and its determinants among adolescent girls in Ethiopia: Evidence from the National Immunization Survey.**
(DTA)

**S1 File. List of abbreviations used in the manuscript.**
(DOCX)

## Acknowledgments

We sincerely thank the University of Gondar, College of Medicine and Health Sciences for their administrative support and for facilitating access to the secondary data used in this study. We are also grateful to the Ministry of Health-Ethiopia, specifically the Maternal, Child, and Adolescent Health Lead Executive Office and the Expanded Program on Immunization (EPI) case team, for their cooperation and for granting permission to access the data. Finally, we acknowledge the participants whose contributions made this research possible.

## Author contributions

**Conceptualization:** Kibir Temesgen Assefa.

**Data curation:** Kibir Temesgen Assefa.

**Formal analysis:** Kibir Temesgen Assefa.

**Funding acquisition:** Kibir Temesgen Assefa.

**Investigation:** Kibir Temesgen Assefa.

**Methodology:** Kibir Temesgen Assefa, Abebaw Gebeyehu Worku, Achenef Asmamaw Muche, Bisrat Misganaw Geremew, Mulat Adefris Woldetsadik, Kassahun Alemu.

**Project administration:** Kibir Temesgen Assefa.

**Resources:** Kibir Temesgen Assefa.

**Software:** Kibir Temesgen Assefa.

**Supervision:** Abebaw Gebeyehu Worku, Achenef Asmamaw Muche, Bisrat Misganaw Geremew, Mulat Adefris Woldetsadik, Kassahun Alemu.

**Validation:** Abebaw Gebeyehu Worku, Achenef Asmamaw Muche, Bisrat Misganaw Geremew, Mulat Adefris Woldetsadik, Kassahun Alemu.

**Visualization:** Kibir Temesgen Assefa.

**Writing – original draft:** Kibir Temesgen Assefa.

**Writing – review & editing:** Abebaw Gebeyehu Worku, Achenef Asmamaw Muche, Bisrat Misganaw Geremew, Mulat Adefris Woldetsadik, Kassahun Alemu.

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
