## [Decision Letter · Decision Letter 0]

11 Mar 2026

PONE-D-25-52998Geographic inequities and determinants of human papillomavirus vaccine non-uptake among adolescent girls in Ethiopia: evidence from national immunization surveyPLOS One

Dear Dr. Assefa,

Thank you for submitting your manuscript to PLOS ONE. After careful consideration, we feel that it has merit but does not fully meet PLOS ONE’s publication criteria as it currently stands. Therefore, we invite you to submit a revised version of the manuscript that addresses the points raised during the review process.

There are only few suggestions and the article may be accepted after your revision.

We look forward to receiving your revised manuscript.

Kind regards,

Ricardo Q. Gurgel, PhD

Academic Editor

PLOS One

“This work is part of PhD study for KTA and was funded by College of Medicine and Health Sciences, University of Gondar with reference number of PGC//098/12/2016.”

“This work is part of PhD study for KTA and was funded by College of Medicine and Health Sciences, University of Gondar with reference number of PGC//098/12/2016.”

“This work is part of PhD study for KTA and was funded by College of Medicine and Health Sciences, University of Gondar with reference number of PGC//098/12/2016.”

5. We note that Figures 3, 4, 5, 6, 7, and 8 in your submission contain [map/satellite] images which may be copyrighted. All PLOS content is published under the Creative Commons Attribution License (CC BY 4.0), which means that the manuscript, images, and Supporting Information files will be freely available online, and any third party is permitted to access, download, copy, distribute, and use these materials in any way, even commercially, with proper attribution. For these reasons, we cannot publish previously copyrighted maps or satellite images created using proprietary data, such as Google software (Google Maps, Street View, and Earth). For more information, see our copyright guidelines: http://journals.plos.org/plosone/s/licenses-and-copyright.

a. You may seek permission from the original copyright holder of Figures 3, 4, 5, 6, 7, and 8 to publish the content specifically under the CC BY 4.0 license.

6. Please include a separate caption for each figure in your manuscript.

Reviewers' comments:

Reviewer's Responses to Questions

**Comments to the Author**

1. Is the manuscript technically sound, and do the data support the conclusions?

Reviewer #1: Yes

Reviewer #2: Yes

2. Has the statistical analysis been performed appropriately and rigorously? 

Reviewer #1: Yes

Reviewer #2: Yes

3. Have the authors made all data underlying the findings in their manuscript fully available?

The PLOS Data policy requires authors to make all data underlying the findings described in their manuscript fully available without restriction, with rare exception (please refer to the Data Availability Statement in the manuscript PDF file). The data should be provided as part of the manuscript or its supporting information, or deposited to a public repository. For example, in addition to summary statistics, the data points behind means, medians and variance measures should be available. If there are restrictions on publicly sharing data—e.g. participant privacy or use of data from a third party—those must be specified.requires authors to make all data underlying the findings described in their manuscript fully available without restriction, with rare exception (please refer to the Data Availability Statement in the manuscript PDF file). The data should be provided as part of the manuscript or its supporting information, or deposited to a public repository. For example, in addition to summary statistics, the data points behind means, medians and variance measures should be available. If there are restrictions on publicly sharing data—e.g. participant privacy or use of data from a third party—those must be specified.requires authors to make all data underlying the findings described in their manuscript fully available without restriction, with rare exception (please refer to the Data Availability Statement in the manuscript PDF file). The data should be provided as part of the manuscript or its supporting information, or deposited to a public repository. For example, in addition to summary statistics, the data points behind means, medians and variance measures should be available. If there are restrictions on publicly sharing data—e.g. participant privacy or use of data from a third party—those must be specified.requires authors to make all data underlying the findings described in their manuscript fully available without restriction, with rare exception (please refer to the Data Availability Statement in the manuscript PDF file). The data should be provided as part of the manuscript or its supporting information, or deposited to a public repository. For example, in addition to summary statistics, the data points behind means, medians and variance measures should be available. If there are restrictions on publicly sharing data—e.g. participant privacy or use of data from a third party—those must be specified.

Reviewer #1: Yes

Reviewer #2: Yes

4. Is the manuscript presented in an intelligible fashion and written in standard English?

Reviewer #1: Yes

Reviewer #2: Yes

5. Review Comments to the Author

Reviewer #1: This paper is very well written. Addresses a very important area to focus on in public health.

I just have minor comments

Human papillomaviruses (HPVs) are a group of epitheliotropic viruses, many of which are

carcinogenic and strongly linked to cervical, anal, oropharyngeal, and other anogenital

cancers.-Kindly include a reference

HPV-related cervical cancer is the world’s most common cancer in women-Add a reference

What are the implications of the findings for public health and policy?

Reviewer #2: The article contains most of the requirements necessary for good publication. It has scientific relevance and was based on strong references. It presents results and conclusions in accordance with the proposed objectives.

I suggest including in the methods section which official reference was used to establish the age of adolescence. In the present study, it cites 14 years or older and it was left vague.

It is known that there are different minimum and maximum age standards. Depending on the location, there are differences between the minimum and maximum age for teenagers. This can modify some data.

Another point to detail is what standard was used to differentiate between levels of wealth and poverty, since this variable was significant in the study.

6. PLOS authors have the option to publish the peer review history of their article (what does this mean?). If published, this will include your full peer review and any attached files.). If published, this will include your full peer review and any attached files.). If published, this will include your full peer review and any attached files.). If published, this will include your full peer review and any attached files.

...

Reviewer #1: No

Reviewer #2: No

---

## [Author Response · Author response to Decision Letter 1]

21 Mar 2026

To Academic Editor

Response: We sincerely thank the editor for this important reminder. After carefully reviewing the PLOS ONE style templates and formatting guidelines, we have revised the manuscript and supporting documents to align with the journal's formatting guidelines, including Proper formatting of headings and sections, Correct file naming conventions and Appropriate structure as per the journal's templates.

“This work is part of PhD study for KTA and was funded by College of Medicine and Health Sciences, University of Gondar with reference number of PGC//098/12/2016.”

Response: Dear editor, thank you for this pointing out this. We now stated the following in the funding statement section:

The funders had no role in study design, data collection and analysis, decision to publish, or preparation of the manuscript. The revision can be found on Page 15, Line numbers 391–393.

“This work is part of PhD study for KTA and was funded by College of Medicine and Health Sciences, University of Gondar with reference number of PGC//098/12/2016.”

Response: We have edited the Acknowledgment section and other areas of the manuscript to avoid funding-related texts. The funding information now appears only in the Funding Statement section of the online submission form. Refer Page 15, Line numbers 376–382.

Response: Thank you. We confirm that we will make all data underlying the findings fully available. We have updated our Data Availability Statement as follows: All relevant data are within the manuscript and its Supporting Information file. The revision can be found on Page 15; line number 388.

5. We note that Figures 3, 4, 5, 6, 7, and 8 in your submission contain [map/satellite] images which may be copyrighted.

Response: Thank you for bringing this important copyright issue to our attention.

The administrative boundary shapefiles used in this study were obtained from the Humanitarian Data Exchange (HDX) platform (https://data.humdata.org/dataset/cod-ab-eth), which provides open-access data publicly available for unrestricted use. Refer Figures 3–8 and corresponding captions.

6. Please include a separate caption for each figure in your manuscript.

Response: Thank you for this reminder. We have ensured that each figure in the manuscript now has its own descriptive caption placed immediately below the figure. The captions provide sufficient detail for readers to understand the figure content without referring to the main text. Please refer figures 1–8 and corresponding captions.

Response: Not applicable because neither of the reviewers recommended citing specific previously published works.

Response: We have thoroughly reviewed our reference list and confirm that all references are complete and correctly formatted according to PLOS ONE style. We did not also cite retracted articles. Refer Pages 16–20; line number 399–556.

Reviewer 1

Overall assessment: This paper is very well written. Addresses a very important area to focus on in public health.

I just have minor comments.

Response: Thank you for your positive assessment of our work. We appreciate your recognition of the importance of this research area. We have carefully addressed each of your minor comments below.

1. Human papillomaviruses (HPVs) are a group of epitheliotropic viruses, many of which are carcinogenic and strongly linked to cervical, anal, oropharyngeal, and other anogenital cancers.-Kindly include a reference.

Response: Thank you for this suggestion. We have added appropriate references to support this statement. Refer Page 3, Line number 63.

2. HPV-related cervical cancer is the world’s most common cancer in women-Add a reference.

Response: Thank you for your observation. We have put specific reference for the indicated statement. Refer Page 3, Line number 64.

3. What are the implications of the findings for public health and policy?

Response: Thank you for raising this important point. We recognize that while we attempted to discuss implications in the concluding sentence of each paragraph throughout the Discussion section, this aspect could be strengthened. In response, we have revised the Discussion to more explicitly address the public health and policy implications of our findings. The revisions can be found on Pages 12–14, Line numbers 286–288, 296–300, 312–315, 326–329, 341–343, and 353–356.

Reviewer 2

Overall: The article contains most of the requirements necessary for good publication. It has scientific relevance and was based on strong references. It presents results and conclusions in accordance with the proposed objectives.

Response: Thank you for your positive assessment of our work. We appreciate your recognition of its scientific relevance and robust methodological approach. We have carefully addressed your specific suggestions below.

1. "I suggest including in the methods section which official reference was used to establish the age of adolescence. In the present study, it cites 14 years or older and it was left vague. It is known that there are different minimum and maximum age standards. Depending on the location, there are differences between the minimum and maximum age for teenagers. This can modify some data."

Response: Thank you for raising this important point. We acknowledge that the initial description of "14 years or older" was unclear; we have now revised it to 15–18 years. This cut-off was determined by the age distribution in the dataset. Although the World Health Organization defines adolescents as individuals aged 10–19 years, our study focuses specifically on girls aged 15–18 years. This was due to the nature of the secondary data, which were originally collected from adolescent girls eligible for HPV vaccination under a single-age cohort strategy targeting only 14-year-olds. As data were collected retrospectively, one year after the vaccination campaign, participants who were 14 at the time of vaccination had turned 15 by the time of data collection. Therefore, 15 years was used as the lower age limit, while the upper age limit (18 years) was determined by the maximum age observed in the dataset. Refer Page 4, Line numbers 108–116.

2. Another point to detail is what standard was used to differentiate between levels of wealth and poverty, since this variable was significant in the study.

Response: Thank you for raising this concern. We agree that clarity regarding the construction of the wealth index is essential for interpreting the findings. We have provided detailed information about the calculation of the wealth index as follows:

The household wealth index was constructed using principal component analysis (PCA). The index incorporated housing characteristics (flooring, roofing, wall materials, water source, toilet facility, and cooking fuel) and asset ownership (television, radio, refrigerator, bicycle, motorcycle, car, mobile telephone, agricultural land, and livestock). PCA generated factor scores, with the first principal component used to create the wealth index, ranking households into quintiles from poorest to wealthiest. For our analysis, we categorized households into five wealth levels (poorest, poorer, middle, richer, and richest), following standard DHS practice to enable cross-study comparisons. The revision can be found on page 6, Line numbers 158–166.

2.

---

## [Decision Letter · Decision Letter 1]

13 Apr 2026

Geographic inequities in human papillomavirus vaccine non-uptake and its determinants among adolescent girls in Ethiopia: Evidence from the National Immunization Survey

PONE-D-25-52998R1

Dear Dr. Assefa,

We’re pleased to inform you that your manuscript has been judged scientifically suitable for publication and will be formally accepted for publication once it meets all outstanding technical requirements.

Kind regards,

Ricardo Q. Gurgel, PhD

Academic Editor

PLOS One

Additional Editor Comments (optional):

Reviewers' comments:

Reviewer's Responses to Questions

**Comments to the Author**

1. If the authors have adequately addressed your comments raised in a previous round of review and you feel that this manuscript is now acceptable for publication, you may indicate that here to bypass the “Comments to the Author” section, enter your conflict of interest statement in the “Confidential to Editor” section, and submit your "Accept" recommendation.

Reviewer #2: All comments have been addressed

2. Is the manuscript technically sound, and do the data support the conclusions?

Reviewer #2: Yes

3. Has the statistical analysis been performed appropriately and rigorously? 

Reviewer #2: Yes

4. Have the authors made all data underlying the findings in their manuscript fully available?

The PLOS Data policy requires authors to make all data underlying the findings described in their manuscript fully available without restriction, with rare exception (please refer to the Data Availability Statement in the manuscript PDF file). The data should be provided as part of the manuscript or its supporting information, or deposited to a public repository. For example, in addition to summary statistics, the data points behind means, medians and variance measures should be available. If there are restrictions on publicly sharing data—e.g. participant privacy or use of data from a third party—those must be specified.requires authors to make all data underlying the findings described in their manuscript fully available without restriction, with rare exception (please refer to the Data Availability Statement in the manuscript PDF file). The data should be provided as part of the manuscript or its supporting information, or deposited to a public repository. For example, in addition to summary statistics, the data points behind means, medians and variance measures should be available. If there are restrictions on publicly sharing data—e.g. participant privacy or use of data from a third party—those must be specified.requires authors to make all data underlying the findings described in their manuscript fully available without restriction, with rare exception (please refer to the Data Availability Statement in the manuscript PDF file). The data should be provided as part of the manuscript or its supporting information, or deposited to a public repository. For example, in addition to summary statistics, the data points behind means, medians and variance measures should be available. If there are restrictions on publicly sharing data—e.g. participant privacy or use of data from a third party—those must be specified.requires authors to make all data underlying the findings described in their manuscript fully available without restriction, with rare exception (please refer to the Data Availability Statement in the manuscript PDF file). The data should be provided as part of the manuscript or its supporting information, or deposited to a public repository. For example, in addition to summary statistics, the data points behind means, medians and variance measures should be available. If there are restrictions on publicly sharing data—e.g. participant privacy or use of data from a third party—those must be specified.

Reviewer #2: Yes

5. Is the manuscript presented in an intelligible fashion and written in standard English?

Reviewer #2: Yes

6. Review Comments to the Author

Reviewer #2: This study is scientifically relevant, particularly in the field of public health, and is a current topic. The authors answered all the questions posed and also accepted the suggestions of the editors and reviewers.

It already had great content and now it's worthy of publication.

7. PLOS authors have the option to publish the peer review history of their article (what does this mean?). If published, this will include your full peer review and any attached files.). If published, this will include your full peer review and any attached files.). If published, this will include your full peer review and any attached files.). If published, this will include your full peer review and any attached files.

...

Reviewer #2: **Yes:** LIGIA MARA DOLCE DE LEMOSLIGIA MARA DOLCE DE LEMOSLIGIA MARA DOLCE DE LEMOSLIGIA MARA DOLCE DE LEMOS

---

## [Editor Report · Acceptance letter]

PONE-D-25-52998R1

PLOS One

Dear Dr. Assefa,

I'm pleased to inform you that your manuscript has been deemed suitable for publication in PLOS One. Congratulations! Your manuscript is now being handed over to our production team.

Kind regards,

on behalf of

Professor Ricardo Q. Gurgel

Academic Editor

PLOS One